# Prediction of Bacteremia Based on 12-Year Medical Data Using a Machine Learning Approach: Effect of Medical Data by Extraction Time

**DOI:** 10.3390/diagnostics12010102

**Published:** 2022-01-03

**Authors:** Kyoung Hwa Lee, Jae June Dong, Subin Kim, Dayeong Kim, Jong Hoon Hyun, Myeong-Hun Chae, Byeong Soo Lee, Young Goo Song

**Affiliations:** 1Division of Infectious Diseases, Department of Internal Medicine, Yonsei University College of Medicine, Seoul 06273, Korea; khlee0309@yuhs.ac (K.H.L.); subink93@yuhs.ac (S.K.); dayoung747@yuhs.ac (D.K.); ayu870213@yuhs.ac (J.H.H.); 2Department of Family Medicine, Yonsei University College of Medicine, Seoul 06273, Korea; s82tonight@yuhs.ac; 3Selvas Artificial Intelligence Incorporate, Seoul 08594, Korea; victor.m.chae@selvas.com (M.-H.C.); jacob.b.lee@selvas.com (B.S.L.)

**Keywords:** bacteremia, prediction, machine learning, data extraction time

## Abstract

Early detection of bacteremia is important to prevent antibiotic abuse. Therefore, we aimed to develop a clinically applicable bacteremia prediction model using machine learning technology. Data from two tertiary medical centers’ electronic medical records during a 12-year-period were extracted. Multi-layer perceptron (MLP), random forest, and gradient boosting algorithms were applied for machine learning analysis. Clinical data within 12 and 24 hours of blood culture were analyzed and compared. Out of 622,771 blood cultures, 38,752 episodes of bacteremia were identified. In MLP with 128 hidden layer nodes, the area under the receiver operating characteristic curve (AUROC) of the prediction performance in 12- and 24-h data models was 0.762 (95% confidence interval (CI); 0.7617–0.7623) and 0.753 (95% CI; 0.7520–0.7529), respectively. AUROC of causative-pathogen subgroup analysis predictive value for *Acinetobacter baumannii* bacteremia was the highest at 0.839 (95% CI; 0.8388–0.8394). Compared to primary bacteremia, AUROC of sepsis caused by pneumonia was highest. Predictive performance of bacteremia was superior in younger age groups. Bacteremia prediction using machine learning technology appeared possible for acute infectious diseases. This model was more suitable especially to pneumonia caused by *Acinetobacter baumannii*. From the 24-h blood culture data, bacteremia was predictable by substituting only the continuously variable values.

## 1. Introduction

Bacteremia is a life-threatening disease that can progress to sepsis and septic shock, finally resulting in death [1,2]. Therefore, it is essential to identify bacteremia early and be able to predict its progression. Early intervention with appropriate antibiotics is important for patients with bacteremia [3,4,5], and differentiation of bacterial and viral infections is essential to prevent antibiotic abuse. By reducing unnecessary exposure to antibiotics in viral infections, the occurrence of antibiotic-related adverse reactions and the emergence of multidrug-resistant organisms can be reduced [6,7].

The field of making prediction models using machine learning has made rapid progress in recent years, and fields such as diagnosis using medical images and analysis of unexpected events using electrocardiogram are particularly active [8,9,10]. As more data is accumulated year-by-year, it is expected that performance of prediction will improve. Meanwhile, although the prediction model for acute infectious diseases is still limited, the ripple effect on public health, such as reduction of infectious-disease-related deaths and appropriate use of antibiotics, will be very inspiring. 

The prediction model of infectious diseases using machine learning has been used particularly for diagnoses in specific clinical settings, such as intensive care units or emergency rooms, and for diagnosing specific bacterial pathogens [11,12,13,14]. So far, machine learning approaches targeting general acute infectious diseases rather than specific infection have not been explored. Thus, we focused on analyzing the risk of progression to bacteremia when blood cultures have been obtained from in-hospital patients. Our previous studies have presented a model for predicting bacteremia using 3-year medical data [15,16]. Based on our experience from the previous study, we extracted a larger clinical dataset over a period longer than 10 years and applied learning and validation processes. 

Thus, we aimed to develop a prediction model for bacteremia using causative bacterial strain, site of infection, age, and sex of the target patients, and propose a predictive model using a machine learning technique that can be comprehensively applied to any clinical situation. In addition, the medical data were divided by extraction time into two groups (within 12 h vs. 24 h of blood culture) and the effect of data extraction time on deep learning analysis was verified.

## 2. Materials and Methods

### 2.1. Study Population 

From 1 January 2007–31 December 2018, a total of 685,225 blood culture test data were extracted from Sinchon and Gangnam Severance Hospitals, Yonsei University-affiliated tertiary-care centers in Seoul, Republic of Korea. During a 12-year period, 521,948 cases from Sinchon Severance Hospital and 163,277 cases from Gangnam Severance Hospital were extracted. All inpatients who underwent blood culture tests, regardless of their admitted location (emergency room, general ward, or intensive care unit), were included in this study. Of these, 62,454 cases in patients younger than 18 years old were excluded from the study, and 560,036 cases with no blood culture growth were included as a control group with non-bacteremia. Only confirmed true bacteremia data were included in bacteremia group; those considered to be contamination of normal skin flora (13,402 cases) and fungal infection (4871 cases) were also excluded from the bacteremia group. The 5710 cases of duplicated blood culture tests within 24 h in the same patient were not included in the bacteremia group. Finally, 38,752 true bacteremia episodes were identified from 622,771 blood culture episodes, and 560,036 non-bacteremia episodes were used as controls (Figure 1). 

### 2.2. Data Extraction

Medical data were extracted from the Clinical Data Warehouse (CDW) system in the Yonsei University Health System. We constructed a training dataset using the result of bacteremia, excluding data without blood culture results. Data duplication for the same patient was avoided by allotting unique identification. If both positive and negative data were present simultaneously for the same patient, the negative data were removed. Fungemia was not included in the training data in any of the positive/negative data samples by being initially excluded from the data set. The negative data were trained using pure non-bacteremia results in blood culture.

All medical data from the two tertiary centers, Sinchon and Gangnam Severance Hospitals, were combined. Each data set consisted of training, validation, and test in the ratio 8:1:1; the training and validation datasets were used for model training, while the test dataset was used for verification. 

### 2.3. Definition of Bacteremia 

Patients with more than one blood culture test within 24 h were considered to have the same episode. Thus, when additional blood culture tests were performed for the same patients after 24 h, it was considered a new episode [17]. The primary endpoint of this study was true bacteremia defined as a positive outcome in all blood culture episodes. For true bacteremia, common contaminants such as *Bacillus* spp., viridans-group streptococci, and coagulase-negative staphylococci, were excluded based on the United States National Healthcare Safety Network system [18,19]. 

However, cases of positive results in two pairs of common contaminants in three other different blood cultures were considered as true bacteremia. In general, blood culture tests were performed when three pairs of tests had been completed on samples collected from three different peripheral vessels, and when one of the three pairs had been identified with common contaminants, it was not regarded as true bacteremia. When bacteremia and non-bacteremia data exist simultaneously in one patient, negative data were deleted, and only positive data were selected for training.

### 2.4. Clinical Variables 

The age, sex, vital signs, and the results of the laboratory test within 24 h from the onset of blood culture time were used as analytical variables. Clinical data within 12 h of blood culture were further analyzed for comparison with the 24-h data group. The clinical variables were analyzed by numeric values, not stratified. Overall, 92 variables were controlled for, and cases missing over 80% of results were excluded during data analysis. Each variable was also modified by a change of annual scale. For example, in Gangnam Severance Hospital, the measurement unit used for C-reactive protein (CRP) data was mg/dL until June 2009, but this was changed to mg/L from July 2009. Thus, the data for CRP estimates before June 2009 were weighted ten times more thereafter. After excluding missing and duplicated data, 34 clinical variables were included for further ranking in the method analysis. The clinical variable of hospital stay refers to the length of hospitalization from the first day of admission to the time for blood culture tests of the study subjects. The inpatient’s total length of hospital stay was not considered as a clinical variable because it was based on the time point when the prediction model could be applied.

### 2.5. Subgroup Analysis

The causative pathogen of bacteremia was analyzed from 12-year medical records, and is shown in Appendix A. Bacteria were also identified in samples other than blood, such as urine, sputum, and bile; the results were used to classify the infection site as urinary tract infection, pneumonia, or biliary tract infection. When the same pathogen as in blood culture was identified in urine, sputum, and bile within 1 week before or after blood culture, it was defined as the infection at the relevant site. When blood culture was positive but there was no bacterial growth in urine, sputum, or bile culture, it was classified as primary bacteremia. The age at the time of blood culture was divided into 18–39, 40–59, and 60–80 years. The analysis was conducted by dividing the group into young and immunologically competent age groups to show gradually decreasing immune function with age. Those over 80 years of age were excluded from the refining process due to accounting for less than 1% of the total data.

### 2.6. Machine Learning Technique and Statistical Analysis

The training data were used to develop a prediction model using the conventional statistical approach and machine learning techniques, the multi-layer perceptron (MLP), random forest, and gradient boosting algorithm (GBM), and then validation was performed [20,21,22,23]. We constructed a standard multi-layer feedforward neural network (MLP with the following three layers: one input, two hidden layers, and one output layer). We applied 128 nodes in the hidden layer. The model was trained to optimize the cross-entropy loss using the Root Mean Square Propagation optimizer with mini-batches of size 64, a learning rate of 0.001, a momentum of 0.9, and a dropout of 0.5 as a regularization technique. Finally, the early stopping technique was applied to prevent overfitting [24,25,26]. For GBM, we chose extreme gradient boosting (XGBoost) as the implementation. Furthermore, we performed the comparison with XGBoost, Gradient Boosting Trees (Gbtree)-based mode that used regression tree as a weak learner, and Dropout meets Additive Regression Trees (DART) [27].

In addition, the performance of these models was verified by comparison with the non-neural network model, random forest. The prediction performance was assessed based on the area under the receiver operating characteristic curve (AUROC), sensitivity, and specificity from the validation data. The resulting AUROCs and the 95% confidence intervals (95% CI) were calculated. We tried to show the influence ranking by the one-out search (OOS) method, which is a technique that learns a model by sequentially excluding the features one-by-one from the variables in a model learning process, with many feature variables used as input. The difference in performance was then calculated, and the ranking of variables was set in an order based on the performance change. Furthermore, we implemented MLP for the subgroup analysis to make the prediction model for the type of pathogen, source of infection, and comparison of prediction accuracy by AUROC in stratified age groups. 

### 2.7. Ethics Approval and Consent to Participate

The protocol for this retrospective study was reviewed and approved by the Institutional Review Board of Gangnam Severance Hospital, Yonsei University College of Medicine in Seoul, Korea (Reg. No. 3-2017-0315, approval date, 5 June 2018). The board waived the requirement for informed consent. All procedures were conducted following the guidelines of the Declaration of Helsinki.

## 3. Results

### 3.1. Comprehensive Analysis

A total of 38,752 bacteremia episodes were identified from 622,771 blood culture episodes. For the 12-year period, the medical data were divided into three categories: training, validation, and test data. The two institutions had a three-fold difference in the size of inpatient admission capacity and a difference in the data disposition as a proportion of disease severity; therefore, the integrated data were trained and applied for internal validation.

When the clinical variables within 12 or 24 h before the blood culture time point were applied, the predicting performance of bacteremia was slightly higher in the 12-h data group (Table 1). In the MLP with 128 hidden layer nodes, AUROC of the prediction performance was 0.762 (95% CI; 0.7617–0.7623) in the 12 h before blood culture time data and 0.753 (95% CI; 0.7520–0.7529) in the 24 h before blood culture time data; in the random forest model, the AUROC was 0.758 (95% CI; 0.7572–0.7591) and 0.738 (95% CI; 0.7383–0.7401), respectively. Two types of ensemble model—XGboost (Gbtree) and XGboost (DART)—were used for the additional analysis; the AUROC was non-inferior compared with other machine learning methods. Furthermore, the AUROC in the 12-h data group was slightly higher than that in the 24-h data group: AUROC, 0.745 (95% CI; 0.7446–0.7455) in the 12-h model vs. 0.730 (95% CI; 0.7300–0.7304) in the 24-h model, using XGboost (Gbtree); 0.744 (95% CI; 0.7439–0.7446) in the 12-h model vs. 0.727 (95% CI; 0.7256–0.7275) in the 24-h model, using XGboost (DART).

### 3.2. Influence Ranking of Clinical Variables

The influence on bacteremia prediction of the clinical variables applied to the machine learning technique analysis was analyzed using the OOS technique; the variables are listed by order of influence in Table 2. The monocyte counts were the most critical clinical variables for prediction of bacteremia in both the 12- and 24-h models. The neutrophil and platelet counts also ranked high, and hospital stay ranked third among the influencing variables in the 12-h model but ranked ninth in the 24-h model. In our previous 3-year retrospective analysis model, alkaline phosphatase (ALP) was the top clinical variable [16], but in this study it ranked tenth and fifth in the 12- and 24-h models, respectively.

### 3.3. Type of Pathogen

The causative strains of bacteremia are listed in the order of frequency of detection, and the top causing pathogen was *Escherichia coli (E. coli)*, followed by *Staphylococcus aureus (S. aureus).* Among the 20 clinical strain rankings listed in Appendix A, additional analysis was performed on the top five clinically meaningful strains (*E coli, S. aureus, Klebsiella pneumonia* (*K. pneumonia), Acinetobacter baumannii* (*A. baumannii),* and *Pseudomonas aeruginosa* (*P. aeruginosa*)). In the subgroup analysis by causative pathogen of bacteremia with the 12-h model, the prediction of *A. baumannii* bacteremia demonstrated the highest predictive value, with an AUROC of 0.839 (95% CI; 0.8388–0.8394), and the AUROC for *E. coli* bacteremia was 0.794 (95% CI; 0.7928–0.7946), as shown in Table 3. Prediction of *S. aureus* bacteremia showed the lowest AUROC of 0.6720 (95% CI; 0.6717–0.6741). In the 24-h model, the prediction of *A. baumannii* bacteremia also showed the highest AUROC of 0.840 (95% CI; 0.8400–0.8407). Prediction for *K. pneumonia* showed an AUROC of 0.778 (95% CI; 0.7777–0.7795), while the AUROC of *S. aureus* bacteremia was 0.737 (95% CI; 0.7354–0.7376) (Figure 2A).

### 3.4. Source of Infection

Compared to primary bacteremia, the expected performance of bacteremia with the same strain reported in sputum (i.e., bacteremia caused by pneumonia) showed the highest AUROC: 0.822 (95% CI; 0.8217–0.8299) in the 12-h model and 0.805 (95% CI; 0.8041–0.8052) in the 24-h model. For biliary infection, an AUROC of 0.775 (95% CI; 0.7742–0.7764) was observed in the 12-h model and 0.792 (95% CI; 0.7910–0.7936) in the 24-h model. In contrast, the prediction for primary bacteremia showed an AUROC of 0.561 (95% CI; 0.5572–0.5651) in the 12-h model and 0.583 (95% CI; 0.5823–0.5855) in the 24-h model (Figure 2B).

### 3.5. Age and Sex Subgroup Analyses

In the analysis of the three stratified age groups—young (18–39), middle (40–59), and older (60–80 years)—the number of participants in each group increased as age increased. However, the prediction performance showed the opposite tendency: the AUROC was 0.781 (95% CI; 0.7780–0.7868) in the younger age group and 0.761 (95% CI; 0.7594–0.7622) in the older age group in the 12-h model. In the 24-h model, the AUROC decreased slightly, at 0.758 (95% CI; 0.7554–0.7648) and 0.723 (95% CI; 0.7221–0.7238) in the younger and older age groups, respectively (Figure 2C). Furthermore, there were no significant differences in the prediction performance by gender (AUROC, 0.763 (95% CI; 0.7630–0.7636) vs. 0.759 (95% CI; 0.7586–0.7600) in the 12-h model; 0.748 (95% CI; 0.7473–0.7481) vs. 0.760 (95% CI; 0.7597–0.7608) in the 24-h model) (Figure 2D). The overall comparison between extraction time of data showed similar AUROC with 0.726 (Figure 2E). 

## 4. Discussion

From the analysis of the electronic medical data over the 12-year period, the MLP model demonstrated the best performance of bacteremia prediction compared to using XGBoost or random forest models. This result is consistent with the known best predictive performance of MLP in our previous research on bacteremia prediction using a three-year cohort [16]. 

To explain the influence of the clinical variables, OOS was applied to the ranking methods, with monocyte, platelet, and neutrophil counts identified as the most significant influential variables. These clinical variables are also related to the first defense mechanism: the innate immunity of the host. The monocytes are involved in phagocytosis, and the neutrophils interact with monocytes; they both coordinate an effective immune response [28,29,30]. The function of platelets in sepsis is already well known: the platelet counts are important laboratory findings in the diagnosis of sepsis and severe sepsis, and show lower index compared to that in non-septic control groups [31]. During infection, pro-inflammatory cytokines activate the coagulation system when endothelial damage occurs, and finally activate the platelets. Additionally, endotoxins released by the bacteria may trigger platelet activation. In bacteremia or during progression to sepsis, fibrin deposition occurs and hypercoagulation continues. In this pathogenesis, platelet counts play a critical role, and could be a marker for prognosis of diseases [32,33,34,35]. 

Thus, these clinically influential variables can play a key role in bacteremia prediction using machine learning analysis. In this study, the clinically influential variables showed excellent predictive performance for the progression to bacteremia when blood was cultured due to fever and under general host conditions, rather than due to specific bacterial strains or host situations as reported previously [11,12,13]. Thus, a prediction model can be used to ultimately prevent progression to sepsis and septic shock. 

In subgroup analysis by causative pathogen, *A. baumannii, E. coli,* and *K. pneumoniae* showed high AUROC for bacteremia prediction. Bacteremia due to pneumonia, biliary, or urinary tract infections showed more prominent predictive values than primary bacteremia. The causative bacterial pathogens of primary bacteremia are almost always nosocomial catheter-related or community-acquired soft tissue skin infection. In particular, *S. aureus* is the most common Gram-positive pathogen of primary bacteremia [36,37]. This is consistent with the result of higher prediction of Gram-negative bacteria than Gram-positive strains in subgroup analysis by causative pathogen in our research. The severity of pneumonia or biliary sepsis was higher than that of primary bacteremia, it was therefore very encouraging to note that the performance of bacteremia prediction of pneumonia and biliary sepsis was superior. By improving the performance of the prediction of subgroup diseases, the more severe diseases that can lead to mortality can be predicted, and early intervention can improve patient prognosis and reduce unnecessary antibiotic use [38]. 

In the age subgroups, the change in laboratory data was lower in the elderly, and the numerical change was more sensitive in the younger age group. In the elderly, fever response is often blunted, and the elevation of the white blood cell counts may also be absent even in the presence of bacteremia [39,40]. Our results proved that the predictive performance in the younger age group had the highest value of the three different age subgroups. For the two different sex subgroups, the differences in the predictive values were not significant. 

In some cases, the AUROC was higher for the 24-h model than the 12-h model. That being said, utilization and comparison are possible in both instances, and it was confirmed that there was no difference in AUROC between the two groups when the 24-h data was divided by the 12-h standard (Figure 2E). Therefore, if data within 24 h of blood culture was applied in the analysis, there would be no difficulty in predicting bacteremia at any time. In most cases, blood cultures are performed when a fever is above 37.7 °C. Thus, the clinical data from the blood culture time can be useful for bacteremia analysis, and the physician can decide whether to use antibiotics for febrile patients based on the laboratory data within 24-h. Prompt intervention with appropriate antibiotics can lead to a better prognosis. Therefore, daily application of the patient’s clinical data allows this predictive model to be used for re-evaluating and making decisions about the patient’s condition. 

There are several studies which showed better performance than our model. However, other studies only focused on patients who had a specific pathogen [11,12], were already diagnosed with SIRS [41], or were treated in the intensive care unit or emergency room [13], while our study tried to make a comprehensive prediction model for general inpatients without limiting to a specific medical situation or pathogen. We tried to create a prediction model of bacteremia that can be comprehensively applied regardless of the place where the blood was collected and the clinical conditions of the study subjects. Therefore, we can apply this predictive model to actual medical practice at any time. In order to increase the accuracy of the control group in the training set, only cases in which no pathogens (including fungi) were found in blood culture were classified as the non-bacteremia group. Ultimately, a comprehensively applicable prediction model was developed, and since the AUROC shows differences according to subgroups (type of pathogen and infection site), it is expected that the performance of prediction will be upgraded if additional medical data and interpretations are added in actual clinical practice.

This study has some limitations. First, this is a retrospective study using electronic medical records extracted from the CDW system. Refining 12-years of clinical data, with vast amounts of information, is challenging but essential. If the real-time prediction model built based on this retrospective analysis is verified with prospective data and continuously enhanced and supplemented, the performance of the bacteremia prediction model can be continuously improved. Second, the results of this study were based on the analysis of integrated data extracted from two tertiary medical institutions. Therefore, external validation is required with data extracted from multiple other centers. Third, it is necessary to match our findings with those of novel sepsis-marker studies, which have recently attracted attention [42,43]. In our 12-year retrospective analysis, serum procalcitonin data were only available for the last 5 years. Therefore, inflammatory cytokines, such as interleukin-6, and tumor necrosis factor-α, along with other novel sepsis biomarkers, such as presepsis, CD64, and soluble triggering receptor expressed on myeloid cells-1 (sTREM-1), can be collected continuously as new clinical variables in order to advance the prediction model.

## 5. Conclusions

In conclusion, prediction of bacteremia using a machine learning technique showed high AUROC, with the possibility of predicting acute infectious diseases. This is a predictive model that can be used regardless of the specific bacterial strain, site of infection, or hospital department. Subgroup analysis by bacterial pathogen and infection site revealed that this model is more relevant in predicting pneumonia caused by *A. baumannii*. In addition, when clinical data were applied to deep learning analysis, no significant difference was observed in the data extraction time within 12- or 24-h blood culture times. Therefore, when clinical data is within 24 h of blood culture, bacteremia can be predicted at any time by substituting only the continuously variable values. Daily application of the patient’s clinical data to the prediction model and re-evaluation of the output from the model can lead to appropriate management of patients. Consequently, real-time monitoring based on vital signs and blood test results may improve clinical prognosis in real-life clinical practice. 

## Figures and Tables

**Figure 1 diagnostics-12-00102-f001:**
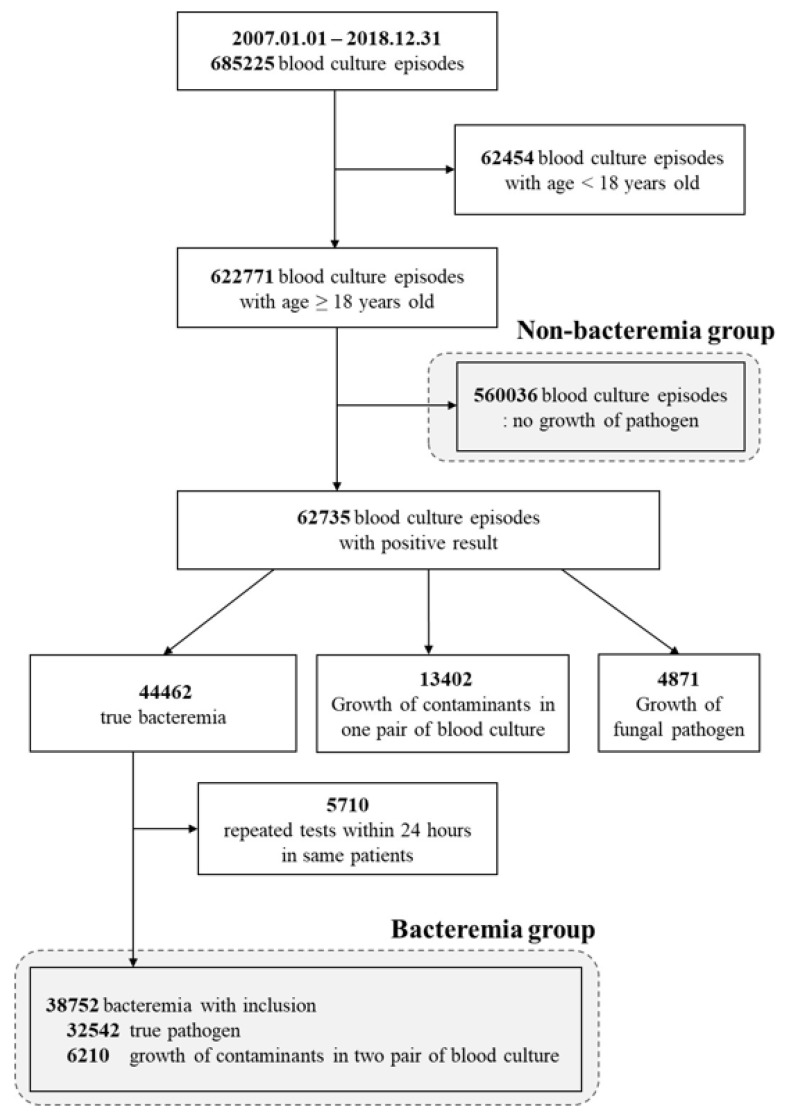
Flow chart of study population.

**Figure 2 diagnostics-12-00102-f002:**
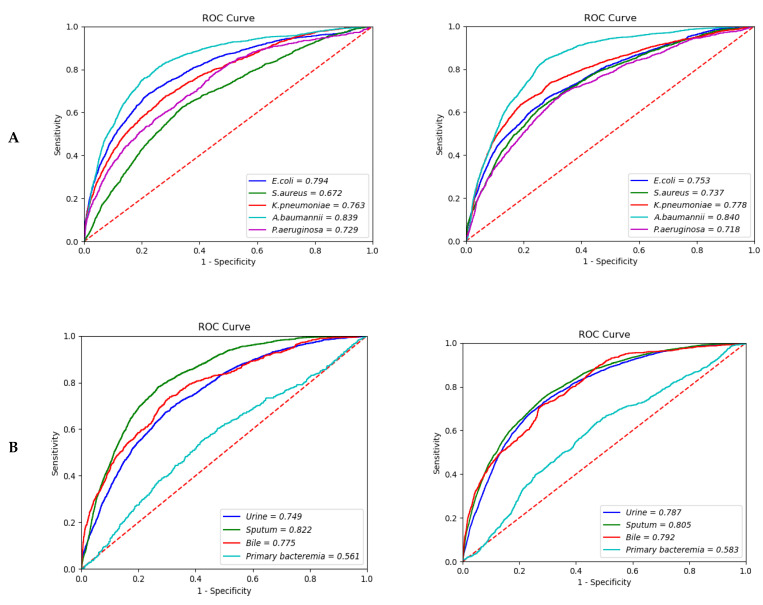
Area under the receiver operating characteristic curve of the bacteremia prediction. (**A**) Type of pathogen (12-h vs. 24-h model); (**B**) Site of infection (12-h vs. 24-h model); (**C**) Age (12-h vs. 24-h model); (**D**) Sex (12-h vs. 24-h model); (**E**) Merge hour (24-h model).

**Table 1 diagnostics-12-00102-t001:** Performance of predicting bacteremia using various machine learning methods.

Type of Data	Model	AUROC (95% CI)	Sensitivity	Specificity
12 h	MLP	0.762 (0.7617–0.7623)	0.695	0.706
	Random Forest	0.758 (0.7572–0.7591)	0.664	0.723
	XGBoost (Gbtree)	0.745 (0.7446–0.7455)	0.629	0.747
	XGBoost (DART)	0.744 (0.7439–0.7446)	0.638	0.747
24 h	MLP	0.753 (0.7520–0.7529)	0.602	0.730
	Random Forest	0.738 (0.7383–0.7401)	0.643	0.729
	XGBoost (Gbtree)	0.730 (0.7300–0.7304)	0.607	0.729
	XGBoost (DART)	0.727 (0.7256–0.7275)	0.602	0.702

Clinical data were extracted within 12 or 24 h of onset of blood culture. Abbreviations: AUROC, area under the receiver operating characteristic curve; CI, confidence interval; MLP, multi-layer perceptron; DART, Dropouts meet multiple Additive Regression Trees.

**Table 2 diagnostics-12-00102-t002:** Influence ranking of clinical variables to bacteremia prediction.

Rank	Data Fusion within 12-h	Data Fusion within 24-h
1	Monocyte	Monocyte
2	Platelet	Neutrophil
3	Hospital stay *	Platelet
4	Neutrophil	Albumin
5	T. bilirubin	ALP
6	BUN	T. bilirubin
7	Albumin	tCO2
8	tCO2	BUN
9	AST	Hospital stay *
10	ALP	CRP
11	ALT	Total Protein
12	White blood cell count	Creatinine
13	Chloride	ALT
14	aPTT	Pulse rate
15	Total Protein	Prothrombin time
16	Pulse rate	Hemoglobin
17	Respiratory rate	AST
18	DBP	Sodium
19	Creatinine	Chloride
20	CRP	ESR

* Hospital stay refers to the length of hospitalization from first day of admission to the time of blood culture tests of the study subjects. Clinical data were integrated based on blood culture time points. Abbreviations: T. bilirubin, total bilirubin; ALP, alkaline phosphatase; BUN, blood urea nitrogen; AST, aspartate transaminase; CRP, C-reactive protein; ALT, alanine transaminase; aPTT, activated partial thromboplastin time; DBP, diastolic blood pressure; ESR, erythrocyte sedimentation rate.

**Table 3 diagnostics-12-00102-t003:** Subgroup analysis of bacteremia prediction according to causing pathogen, infection site, age, and sex.

Type of Data	Subgroup		With Bacteremia	Without Bacteremia	AUROC (95% CI)	Sensitivity	Specificity
12 h	Pathogen	*E. coli*	1805	14,068	0.794 (0.7928–0.7946)	0.693	0.766
		*S. aureus*	1827	14,068	0.672 (0.6717–0.6741)	0.656	0.618
		*K. pneumoniae*	1518	14,068	0.763 (0.7616–0.7658)	0.677	0.716
		*A. baumannii*	1727	14,068	0.839 (0.8388–0.8394)	0.789	0.750
		*P. aeruginosa*	855	14,068	0.729 (0.7278–0.7331)	0.611	0.706
	Infection site	Urine	2202	21,540	0.749 (0.7485–0.7504)	0.642	0.725
		Sputum	3051	21,540	0.822 (0.8217–0.8229)	0.792	0.715
		Bile	650	21,540	0.775 (0.7742–0.7764)	0.739	0.684
		Primary bacteremia	557	21,540	0.561 (0.5572–0.5651)	0.636	0.473
	Age	18–39 years	1377	3885	0.781 (0.7780–0.7868)	0.3923	0.8898
		40–59 years	5141	8359	0.718 (0.7152–0.7190)	0.5818	0.7382
		60–80 years	8936	14,862	0.761 (0.7594–0.7622)	0.7405	0.6597
	Sex	Male	9416	117,535	0.763 (0.7630–0.7636)	0.710	0.694
		Female	6038	81,236	0.759 (0.7586–0.7600)	0.670	0.724
24 h	Pathogen	*E. coli*	2771	22,114	0.753 (0.7523–0.7353)	0.639	0.738
		*S. aureus*	2949	22,114	0.737 (0.7354–0.7376)	0.668	0.684
		*K. pneumoniae*	2376	22,114	0.778 (0.7777–0.7795)	0.706	0.730
		*A. baumannii*	2621	22,114	0.840 (0.8400–0.8407)	0.817	0.747
		*P. aeruginosa*	1280	22,114	0.718 (0.7170–0.7209)	0.688	0.661
	Infection site	Urine	3489	34,502	0.787 (0.7860–0.7878)	0.751	0.689
		Sputum	4700	34,502	0.805 (0.8041–0.8052)	0.670	0.756
		Bile	1022	34,502	0.792 (0.7910–0.7936)	0.659	0.740
		Primary bacteremia	987	7169	0.583 (0.5823–0.5855)	0.517	0.616
	Age	18–39 years	413	5712	0.758 (0.7554–0.7648)	0.463	0.845
		40–59 years	1415	12,415	0.719 (0.7176–0.7214)	0.566	0.749
		60–80 years	3026	21,154	0.723 (0.7221–0.7238)	0.560	0.713
	Sex	Male	14,585	241,478	0.748 (0.7473–0.7481)	0.673	0.707
		Female	10,030	164,521	0.760 (0.7597–0.7608)	0.619	0.760
	Merge hour	Under 12 h	323,949	20,608	0.726 (0.7261–0.7266)	0.631	0.705
		Over 12 h	82,050	3957	0.726 (0.7253–0. 7258)	0.653	0.679

Clinical data were integrated based on blood culture time points. Abbreviations: AUC, area under the receiver operating characteristic curve; CI, confidence interval; *E. coli, Escherichia coli*; *S. aureus, Staphylococcus aureus*; *K. pneumoniae, Klebsiella pneumonia*; *A. baumannii, Acinetobacter baumannii*; *P. aeruginosa, Pseudomonas aeruginosa*.

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
