# Peer review of "Prediction of Bacteremia Based on 12-Year Medical Data Using a Machine Learning Approach: Effect of Medical Data by Extraction Time"

_diagnostics, 2022, doi:10.3390/diagnostics12010102_

Round 1

Reviewer 1 Report

Lee, et al performed a machine learning method to predict bacteremia which is important to physicians in clinical practice. The study population is large, and the idea is good. However, the biggest concern is the study design. Maybe it is my problem, but I’m confused about the main aim of this study. The authors try to predict bacteremia, but all negative controls also had bacteremia. I don’t’ think the authors could make such conclusions from their current study methods and revision is needed.

The followings are some of my opinions and doubts,

  1. As the authors mentioned in Introduction, they tried to predict bacteremia in any clinical situation (line 50-53). However, only patients with confirmed bacteremia were included for model establishment (line 63-64). Since all included patients had bacteremia, what did author try to differentiate bacteremia from? (From healthy people or sick patients?)
  2. While applying the results to real world, we concern that are our patients so different to those in the study that the results cannot apply? So, authors should clarify that where and why did these included patients have the test for bacteremia? (From emergency department, outpatient department or inpatient department? Because of fever, abdominal pain, or respiratory tract infection?)
  3. Line 111-113. If the authors defined patients without other bacterial growth in other samples as negative control (primary bacteremia), they were establishing models to predict the infection site or infection source, not predicting bacteremia. It is different from their primary endpoint mentioned in line 73-74.
  4. In results, line 156-168 and Table 1., the AUROC was used to evaluate the predicting performance. The authors showed the performance of predicting bacteremia using various machine learning methods. However, all included patients had bacteremia as mentioned in methods (line 63), what exactly did authors try to predict?
  5. Line 194-197 and Table 3. Who are those without bacteremia?
  6. Line 217-219, the prediction for primary bacteremia showed an AUROC of 0.561… What did authors differentiate primary bacteremia from? Were they not negative controls?

Reviewer 2 Report

This article is about predicting bacteremia using machine learning techniques. And the dataset is quite huge and two different tertiary hospital data were used in this analysis. However, there are some methodologic problems in this article.  

  1. There are several studies predicting bacteremia using deep learning approaches. But your introduction only has a few articles about that. More previous articles should be added to this introduction.
  2. How many hidden layers were used in your model? The model described like 1 layer with 128 hidden nodes. If then, the MLP can work only like a usual multivariable model, not a deep learning model.
  3. All the variables used in your model should be described with supplement materials. And I think the vital sign is a major factor that can predict bacteremia. You can easily find the article whose model uses the vital signs and have higher performance than this model. Your model only used the static variables around the bacteremia event. So the model performance is not high enough and the performance of the model is much lower than previously published models.
  4. How did you treat the fungus-positive sample? Is that used as a negative sample or omitted from the whole sample? As the feature of fungal positive data and bacteremia data could be quite similar, it could make the model performance lower because of pseudo negative samples. Moreover, it should be described in the method section.
  5. In table 3, how the infection sites are defined? Did you define a source of infection as the positive culture other than blood? If then you should describe, how the data is handled, for example, positive sputum culture prior several times to bacteremia and how could your definition is reliable.
  6. In result 3.5, you are describing the difference of ROC in different groups. The description of analyzing the confidence interval of the ROC curve is not described in the method section. And the significant difference between subgroups should be based on the roc test in general. As the data shows that the ROC curve of the gender group is quite similar that additional tests could be omitted, but the description of making confidence interval should be described in the method section.
  7. In paragraph 4 of the discussion section, the word “Sepsis due to disease” should be used with caution. In the Methods section, the route of infection is not clearly defined, and it is difficult to define the route of infection in certain situations without a manual chart review. So there is a leap in logic.

Minor comment

Your citation of 9th is miscited. It is not about the artificial intelligence used in diagnosing bacteremia.

Round 2

Reviewer 1 Report

The authors have revised their manuscript and clarified the study group (bacteremia group and control group) clearly. There are some suggestions as below.

1.In the authors’ response, they mentioned” All patients who performed blood culture regardless of emergency room, general ward, or Intensive care unit, the all inpatient were included.” Authors should add this point to their methods to make readers understand the detail of the study group.

2.The authors used “hospital stay” (Table 2.) as a clinical variable to predict bacteremia. I think the hospital stay is a result of bacteremia. Because patient with bacteremia may have longer hospital stay to treat the bacteremia. If the authors mean longer hospital stay before the blood sample collected for culture, they should clarify it to avoid misunderstanding. Maybe use the term “hospital stay before test”.

3.Line 304-309. The authors tried to build a prediction model fits all condition for all inpatient cases (ER, general ward, or ICU). However, the readers should interpret this with caution. Patients from different departments have different risk of bacterial infection. For example, ICU patients have higher risk of nosocomial infection, but ER patients don’t. Intubated ICU patients also have higher risk of pneumonia related bacteremia, but ER patients don’t. The authors should remind readers about this point when they apply this model in clinical practice.

Reviewer 2 Report

This article is about predicting bacteremia using machine learning techniques. And the dataset is quite huge and two different tertiary hospital data were used in this analysis. However, there are some methodologic problems in this article.  

  1. The MLP should consist of multiple hidden layers. If you made a model with one hidden layer, it cannot be the “multi-layer” perceptron. It is a uni-layer perceptron and it cannot represent the deep learning model. As the one-layer model cannot solve the XOR problem, the one-layer model cannot be called “a deep learning model”.( Zhao Y, Deng B, Wang Z. Analysis and study of perceptron to solve XOR problem. Proc - 2nd Int Work Auton Decentralized Syst IWADS 2002. 2002; 168–173.) You should remove the model in your comparison table or you should re-build the model with a more complex one.
  2. In the method section, the fungemia patients were excluded from your dataset. It means that you are omitting the subpopulation with characteristics from the training and validation set. This makes the model’s results less reliable when testing in real-world condition. In this respect, the conclusion cannot be like “real-time prediction in real-world”. This is because clinicians cannot rule out the situation when fungemia is suspected in real practice. And the AUORC is not high enough for real world use. Therefore, the conclusion should be limited and it is recommended to modify the conclusion to focus on the subgroup trait to which the trait differ between pathogen.
